# Bio-Guided Fractionation of Prenylated Benzaldehyde Derivatives as Potent Antimicrobial and Antibiofilm from *Ammi majus* L. Fruits-Associated *Aspergillus amstelodami*

**DOI:** 10.3390/molecules24224118

**Published:** 2019-11-14

**Authors:** Noha Fathallah, Marwa M. Raafat, Marwa Y. Issa, Marwa M. Abdel-Aziz, Mokhtar Bishr, Mostafa A. Abdelkawy, Osama Salama

**Affiliations:** 1Pharmacognosy and Medicinal Plants Department, Faculty of Pharmaceutical Sciences and Pharmaceutical Industries, Future University in Egypt, Cairo 11835, Egypt; noha.mostafa@fue.edu.eg (N.F.); osalama@fue.edu.eg (O.S.); 2Microbiology and Immunology Department, Faculty of Pharmaceutical Sciences and Pharmaceutical Industries, Future University in Egypt, Cairo 11835, Egypt; 3Pharmacognosy Department, Faculty of Pharmacy, Cairo University, Cairo 11562, Egypt; marwa.issa@pharma.cu.edu.eg (M.Y.I.); Gouda48@gmail.com (M.A.A.); 4Regional Center for Mycology and Biotechnology (RCMB), Al-Azhar University, Cairo 11651, Egypt; marwaemam.17@azhar.edu.eg or; 5Arab Company for Pharmaceuticals and Medicinal Plants, El-Sharkya 11361, Egypt; mbishr_2000@yahoo.com

**Keywords:** *Ammi majus*, *Aspergillus amstelodami*, endophytic fungi, molecular identification, antimicrobial activity, anti-biofilm activity, prenylated benzaldehyde derivatives, dihydroauroglaucin

## Abstract

*Ammi majus* L.; Family Apiaceae; is a plant indigenous to Egypt. Its fruits contain bioactive compounds such as furanocoumarins and flavonoids of important biological activities. An endophytic fungus was isolated from the fruits and identified as *Aspergillus amstelodami* (MK215708) by morphology, microscopical characterization, and molecular identification. To our knowledge this is the first time an endophytic fungus has been isolated from the fruits. The antimicrobial activity of the *Ammi majus* ethanol fruits extract (AME) and fungal ethyl acetate extract (FEA) were investigated, where the FEA showed higher antimicrobial activity, against all the tested standard strains. Phytochemical investigation of the FEA extract yielded five prenylated benzaldehyde derivative compounds isolated for the first time from this species: Dihydroauroglaucin (1), tetrahydroauroglaucin (2), 2-(3,6-dihydroxyhepta-1,4-dien-1-yl)-3,6-dihydroxy-5-(dimethylallyl)benzaldehyde (3), isotetrahydroauroglaucin )4), and flavoglaucin (5). Structure elucidation was carried out using (1H- and 13C-NMR). Fractions and the major isolated compound **1** were evaluated for their antimicrobial and antibiofilm activity. Compound **1** showed high antimicrobial activity against *Escherichia coli* with minimum inhibitory concentration (MIC) = 1.95 µg/mL, *Streptococcus mutans* (MIC = 1.95 µg/mL), and *Staphylococcus aureus* (MIC = 3.9 µg/mL). It exhibited high antibiofilm activity with minimum biofilm inhibitory concentration (MBIC) = 7.81 µg/mL against *Staphylococcus aureus* and *Escherichia coli* biofilms and MBIC = 15.63 µg/mL against *Streptococcus mutans* and *Candida albicans* and moderate activity (MBIC = 31.25 µg/mL) against *Pseudomonas aeruginosa* biofilm. This reveals that dihydroauroglaucin, a prenylated benzaldehyde derivative, has a broad spectrum antimicrobial activity. In conclusion, it was observed that the MICs of the FEA are much lower than that of the AME against all susceptible strains, confirming that the antimicrobial activity of *Ammi majus* may be due to the ability of its endophytic fungi to produce effective secondary metabolites.

## 1. Introduction

*Ammi majus (A. majus)* L. of the family Umbellifereae/Apiaceae is an annual herbaceous plant which spreads in Egypt in the Nile and delta region [1], especially in Fayoum and Behira [2]. It is distributed in the Mediterranean region of Europe, western Asia, and now cultivated in India. It is known to contain biologically active compounds such as coumarins and flavonoids [3]. It is usually utilized for the treatment of skin conditions such as psoriasis and vitiligo (acquired leukoderma) [4]. In folk medicine it is used for treatment of urinary tract infections, kidney stones, and leprosy, and as a diuretic and emmenagogue to regulate menstruation [5].

Herbal medicine is considered as one of the earlier applicants of recent pharmaceuticals, used for curing certain diseases. Plant-derived compounds are proposed as alternatives to traditional antimicrobials. Essential oils specially are receiving growing interest as antimicrobial agents that retain high biocompatibility [6,7].

Since symbiosis’s first description in the 19th century [8] as the living together of dissimilar organisms’ many symbiotic lifestyles have been defined based on benefits and impacts to the hosts and the symbionts. About 100 years of research revealed that most, if not all, plants in natural ecosystems are symbiotic with fungal endophytes so that endophytic fungi are located in nearly all plants including herbaceous plants, grass, trees, and algae [9].

Most endophytic fungi are members of the Ascomycetes and fungi imperfecti. Under ordinary conditions they live inside the host plant without producing any indications of illness. Moreover, endophytic fungi are not counted as saprophytes since they are related to living tissues and may participate in the wellbeing of the plant [10]. These fungi have been identified as useful sources of bioactive secondary metabolites and many of them have the ability to produce antimicrobial substances [11,12,13,14].

Worldwide issues caused by drug-resistant bacteria and fungi are considered a health threat. Biofilm forming microorganisms are among the challenges facing scientists nowadays with their distinctive ability to alter their immediate environments by an interesting phenotypical plasticity that involves changes in their physiology and their resistance to antimicrobial agents [15]. Biofilm forming microorganisms are implicated in many infective diseases such as otitis media, periodontitis, dental caries, and osteo-myelitis and chronic diseases such as pulmonary infections of cystic fibrosis patients [16]. Different microorganism strains that resist antibiotics and other dangerous medical conditions demand the search for novel and effective chemical agents from hidden niches. Endophytes are considered as an area of research which can be a promising source for new drug discovery [17]. *A. majus* fruit extract was previously reported to possess antimicrobial activity [2,18]. Consequently, the current study aimed to isolate endophytic fungi associated with *A. majus* fruits and track the antimicrobial and antibiofilm activity using bio-guided fractionation method, which integrates analytical procedures with bioassays, representing a rapid and cost-effective method to discover potential useful fractions and pure compounds [19].

## 2. Results

### 2.1. Isolation and Identification of the Endophytic Fungus

In the present study, a fungus was isolated by repeated culturing of the crushed fruits of *A. majus*. The isolated fungus belonged to *Aspergillus* genus depending on the morphological and microscopical characteristics described in Table 1 and Figure 1. The identification was confirmed using amplification and sequencing of internal transcribed spacer ribosomal RNA (ITS rRNA) gene. Sequence analysis revealed 99% identity with *Aspergillus amstelodami (A. amstelodami)*
Figure 2 and Appendix A. The ITS rRNA gene sequence was submitted at the GenBank under the accession number (MK215708) [20].

### 2.2. Identification and Elucidation of the Isolated Compounds

Fractionation of fungal ethyl acetate (FEA) extract resulted in production of three fractions, fraction I was eluted by *n*-hexane-ethyl acetate (70:30) and appeared as a dark brown band on thin layer chromatography (TLC) (60 mg), fraction II eluted by *n*-hexane-ethyl acetate (60:40) and appeared as yellow band on TLC (100 mg), and finally fraction III eluted with *n*-hexane-ethyl acetate (40:60), it appeared as red band on TLC (79 mg). Compounds **2** (10 mg) and **5** (2.5 mg) were obtained from fraction I, compounds **3** (1 mg), and **4** (1.2 mg) were acquired from fraction II, and the major compound **1** (12 mg) was isolated from fraction III along with two minor compounds that were neglected. The isolated compounds were identified based on their physical characters, spectral data analysis (^1^H-, ^13^C-NMR, and MS) (Appendix A) and by comparing with the literature as seen in Table 2 and Figure 3. 

Compound **1** was the major compound (12 mg) obtained as reddish-brown amorphous powder. It showed attached proton test (APT) spectrum revealing the presence of carbon signals classified as five quaternary, eight methine, three methylenes, and three methyl carbons, the spectrum showed signals at the olefinic region in the heptyl side chain. The proton and ^1^H–^1^H COSY confirmed these signals with (*δ* 6.59, d, *J* = 15.8 Hz, 1H, H-1′) correlated with (*δ* 6.46, m, 1H, H-2′) which is correlated with (*δ* 6.3, m, 1H, H-3′), H-3′ showed correlations with the deshielded methylene at (*δ* 5.47, t, 1H, H-4′) correlated with protons (*δ* 2.1, m, 2H, H-5′), and (*δ* 1.3, m, 2H, H-6′) which is finally correlated with the deshielded terminal CH_3_ group (*δ* 0.9, m, 3H, H-7′). The spectrum confirmed the prenylated benzaldehyde moiety with ortho substitution and two distinct signals at (*δ* 10.1, s, H-7) and (*δ* 11.75, s, H-6). The appearance of dimethylallyl isoprene unit was seen at (*δ* 3.34, d, *J* = 7.8 Hz, 1H, H-1′′) correlated to the olefinic proton (*δ* 5.3, m, 1H, H-2′′) together with two singlet deshielded CH_3_ protons (1.7, s, 3H, H-4′′) and (*δ* 1.8, s, 3H, H-5′′). The presence of a singlet signal at (*δ* 7.0, s, 1H, H-4) indicating aromatic proton. 2D HMBC (Heteronuclear Multiple Bond Correlation) indicated the accurate positions of the olefinic bonds in the side chain and the relation to the aromatic ring, where (*δ* 6.59, d, *J* = 15.8 Hz, 1H, H-1′) showed HMBC relation with the hydroxylated aromatic carbon (*δ*_c_, 145.5, C-3), (*δ*_c_ 130.8, C-1), and (*δ*_c_ 127.5, C-2) which confirms the presence of the unsaturation protons at H-1′ and its relation to C-2 in the aromatic ring. The NMR data of compound **1** were identical with the published for dihydroauroglaucin with molecular formula C_19_H_24_O_3_ [21,22,23].

Compound **2** was isolated as yellowish-brown amorphous powder. The ^1^H NMR confirmed the presence of a similar nucleus as found in compound **1** with some differences at the heptyl side chain. The APT spectrum revealed the presence of 19 carbon signals including six C, five CH, five CH_2_, and three CH_3_, the spectrum showed similarities to compound **1** with some differences at the heptyl side chain as it showed aliphatic CH_2_ groups from C-3′ to C-7′. The ^1^H–^1^H COSY confirmed these changes at (*δ* 6.50, d, *J* = 16.1 Hz, 1H, H-1′) correlated with (*δ* 6.0, m, 1H, H-2′) and (*δ* 2.3, m, 2H, H-3′), respectively. The spectrum confirmed the correlations between H-3′ with H-4′, H-4′with H-5′, H-5′ with H-6′, and finally H-6′ with the methyl group H-7′. HMBC spectrum allowed the positioning of the double bond in the heptyl side chain (*δ* 6.50, d, *J* = 16.1 Hz, 1H, H-1′) showed signals at (*δ*_c_ 144.8, C-3), (*δ*_c_ 117.5, C-1), and (*δ*_c_ 124.5, C-2) which indicates the presence the double bond at H-1′ and its connection to C-2 in the aromatic ring. The NMR data were compared with the literature [24] and the compound was identified as tetrahydroauroglaucin with molecular formula C_19_H_26_O_3._

Compound **3** was isolated as yellow amorphous powder. Its ^1^H NMR, revealed the presence of hydroxybenzaldehyde nucleus as revealed by the chelating hydroxyl group (*δ* 11.95, s, OH, H-6) and the aldehyde proton (*δ* 10.3, s, H-7). Additional evidence of the presence of isoprenyl moiety was proved by the presence of dimethylallyl unit with a methylene proton (*δ* 3.32, d, *J* = 7.4 Hz, 2H, H-1′′) and an olefinic proton (*δ* 5.3, m, H-2′′), two singlet deshielded methyl signals (*δ* 1.7, s, 3H, H-4′’) and (*δ* 1.8, s, 3H, H-5′’). Moreover, the spectra confirmed the unsaturation in heptyl side chain at (*δ* 6.92, d, *J* = 5.5 Hz, 1H, H-1′) and (*δ* 6.0, m, 1H, H-2′). The CH_3_ group at the end of the chain appeared at (*δ* 1.3, m, 3H, H-7′). The aromatic area at the spectrum revealed a singlet aromatic proton (*δ* 7.02, s, 1H, H-4). The presence of peaks at *δ* 7.5 and *δ* 6.4 suggested the presence of hydroxyl groups and by comparing the NMR data of compound **3** with those of the reported data for prenylated benzaldehyde derivatives (Auroglaucin derivatives) [23,25] compound **3** was identified as 2-(-3,6-dihydroxyhepta-1,4-dien-1-yl)-3,6-dihydroxy-5-(dimethylallyl)benzaldehyde with the molecular formula C_19_H_24_O_5_.

Compound **4** was obtained as yellow powder showed ^1^H NMR spectrum similar to that of compound 1 indicating the same main nucleus with differences at H-1′, 2′, 3′, 4′ at (*δ* 2.99, m, 2H, H-1′), (*δ* 1.6, m, 2H, H-2′), (*δ* 1.4, m, 2H, H-3′), (*δ* 1.3, m, 1H, H-4′) indicating the saturation at the heptyl side chain with the double bond present between (*δ* 5.3, m, 1H, H-5′), and (*δ* 5.4, m, 1H, H-6′). The spectrum was compared with published data [23,25,26,27] and with the data obtained from compounds **1**, **2** and **3**, then **4** was identified as Isotetrahydroauroglucin with molecular formula C_19_H_26_O_3_.

Compound **5** was acquired as yellowish powder, its APT spectrum showed the signals of 19 carbons classified as (six quaternary chiral carbons C, three methine CH, seven methylene CH_2_, and three methyl carbons CH_3_, the spectrum revealed some similarities to compounds **1** and **2**, with different signals replacing the olefinic methines with chemical shift in the aliphatic chemical shift area of the saturated heptyl side chain. ^1^H-NMR showed the presence of singlet aromatic proton (*δ* 6.9, s, 1H, H-4) which indicates the substitution on the aromatic ring, the dimethylallyl isoprene unit with one methylene proton at (*δ* 3.31, d, *J* = 7.5 Hz, 1H, H-1′′) and one olefinic proton at (*δ* 5.3, m, 1H, H-2′′) together with two singlet de-shielded methyl protons (*δ*1.7, s, 3H, H-4′′) and (*δ*1.8, s, 3H, H-5′′). Finally, the spectrum preserved the prenylated ortho hydroxy benzaldehyde moiety with two signals for (*δ* 10.3, s, H-7) and (*δ* 11.95, s, OH, H-6). By reviewing the spectrums and by comparing with the published data [20,24,28,29,30,31] it was concluded that compound **5** is Flavoglaucin with the molecular formula C_19_H_28_O_3_.

### 2.3. Antimicrobial Activity Screening

Total *A. majus* ethanol fruits extract (AME) and FEA extract were investigated for their antibacterial and antifungal activities. The FEA extract showed a potent broad spectrum antibacterial and antifungal activities against all the tested strains with MIC values ranging from 6.25 µg/mL to 50 µg/mL, while AME extract showed moderate activity against *Staphylococcus aureus* (*S. aureus), Escherichia coli (E.coli), Streptococcus mutans (S. mutans)*, and *Candida albicans* (*C. albicans)* with minimum inhibitory concentration (MIC) values 25, 25, 50, and 50 µg/mL, respectively, and diminished activity against *Pseudomonas aeruginosa (P. aeruginosa) and Aspergillus fumigates (A. fumigates)* (MIC >100 µg/mL) Figure 4a & Appendix A. The maximum activity of the FEA extract was observed against *S. aureus* and *E. coli* (MIC = 6.25 µg/mL for both). The activity decreased against *S. mutans* and *C. albicans* by one-fold (MIC = 12.5 µg/mL) followed by *P. aeruginosa and A. fumigatus* (MIC = 50 µg/mL). 

Fractionation was done for further and deeper investigation of the most active fraction in the fungus extract as described above. Fraction III showed the strongest antimicrobial activity compared to the other two fractions with MIC values ranging from (3.9 µg/mL) against *S. mutans* to (15.63 µg/mL) against *P. aeruginosa* and *C. albicans*. Fraction I and Fraction II exhibited similar pattern of activity against *S. mutans* (MIC = 31.25 µg/mL)*, C. albicans* (MIC = 31.25 µg/mL), and *P. aeruginosa* (MIC = 62.5 µg/mL), however, fraction I revealed a four times increase in the activity against *S. aureus* (MIC = 15.63 µg/mL) and two times higher activity against *E. coli* (MIC *=* 7.81 µg/mL) when compared to Fraction II as shown in Figure 4b and Appendix A. Dihydroauroglaucin, the major compound isolated from Fraction III, was the most active among all the previously tested extracts and fractions showing potent antimicrobial effect against all evaluated strains with MIC values ranging from (1.95 µg/mL) against *E. coli* and *S. mutans* to (7.81 µg/mL) against *P. aeruginosa*
Figure 4b and Appendix A. All MIC values for antimicrobial activities of AME extract, FEA extract, FEA fractions and compound **1** presented in Appendix A. 

### 2.4. Antibiofilm Assay

The ability of the AME and FEA to disrupt the 24 h preformed biofilm and affect their adhesion was evaluated against reference biofilm-forming strains. It was observed that FEA showed significantly higher inhibition of the biofilm (*p*-value ≤ 0.0087) in comparison to AME (Figure 5a and Appendix A). Furthermore, the three fractions from FEA were assayed and fraction III exhibited the best results when compared to fraction I and fraction II. The highest activity was against *E. coli* with minimum biofilm inhibitory concentration (MBIC) = 15.63 µg/mL followed by the two Gram positive bacteria; *S. aureus* and *S. mutans* with MBIC of 31.25 µg/mL and moderate activity against *P. aeruginosa* biofilm with MBIC of 125 µg/mL (Figure 5b and Appendix A). Fraction I and fraction II showed weak activities against all microorganisms’ biofilm with MBIC ranging from 62.5 µg/mL to 500 µg/mL (Figure 5b and Appendix A).

Depending on the previous results compound **1**; the major isolated compound from fraction III; was selected and tested for its ability to affect the established biofilm. It was found that this compound inhibited biofilm in all types of tested microorganisms with MBIC much lower than that of the total fungal extract and the three fractions. The highest activity was against *S. aureus* and *E. coli* biofilm (MBIC = 7.81 µg/mL) then *S. mutans* and *C. albicans* biofilms (MBIC = 15.63 µg/mL). Likewise, compound **1** revealed an antibiofilm activity against *P. aeruginosa* (MBIC of 31.25 µg/mL) (Figure 5b and Appendix A). All MIC values for antibiofilm activities of AME extract, FEA extract, FEA fractions and compound **1** presented in Appendix A.

### 2.5. Cytocompatibility 

The cytocompatibility effect of AME, FEA, and dihydrauroglaucin on normal human fibroblastic cell line (BHK) was investigated; it was found that AME and FEA did not exhibit toxicity towards BHK as compared with control. FEA exhibited 12% cell death while AME showed about 5% normal cells death only and dihydrauroglaucin caused ca. 40% normal cells killing with the maximum applied concentration (50 μg/mL) (Figure 6). 

## 3. Discussion

Multi-drug resistant microorganisms are a serious problem nowadays resulting in high mortality rate with accompanied community and nosocomial infections. Various approaches to discover new compounds either with antimicrobial activity or enhancing the activity of the present antimicrobial agents were aimed. Searching for compounds from natural microbial sources was our goal. *A. majus* is an indigenous plant in Nile Delta regions of Egypt. It has been studied extensively for its biological antimicrobial activities and was used traditionally for hundreds of years as a treatment for microbial infections [2,32,33,34].

In 2017, Adham and Abdulah, reported the antibacterial and antibiofilm activities of *A. majus* fruits extract against only Gram-positive bacteria [18]. This gave us a clue to design a study using bio-guided fractionation method to specify the compounds responsible for antimicrobial and antibiofilm activity in addition to the investigation of the role of endophytic fungi to the previously reported activity of *A. majus*. One of the most chief characteristics of endophytic microorganisms, particularly fungi, is related to their ability to generate diversity of bioactive molecules that can defend the plant against pathogens [35]. In this study, *A. amstelodami* was isolated as endophytic fungi and to the best of our knowledge; this is the first report on the activity of *A. amstelodami* extract isolated from the *A. majus* fruits. However, endophytic fungi from genus *Aspergillus* have been stated as a source of bioactive secondary metabolites having diverse applications [36,37,38]. Bioactive metabolites derived from endophytic fungi revealed antimicrobial activity against a variety of organisms [13,14].

Solid state fermentation using complex solid rice medium was used in this study as standard condition for large scale production and isolation of sufficient quantities of the compounds of interest. Vander Molen et al. [39] demonstrated that the use of cultures grown on solid media produced extracts with one to two folds larger masses than growing the same fungus in liquid media. The present work evaluates the antimicrobial and antibiofilm activities of the endophytic FEA and compares it to AME. Both extracts showed broad spectrum antibacterial and antifungal activities. Endophytic fungi have close association with host plants and can yield metabolites that encourage vegetative growth and protect the plant against herbivores and microbes [40]. This may rationalize that FEA was found to have higher antibacterial and antifungal effects in addition to stronger inhibition of the preformed biofilm than the total fruit extract AME.

It was observed that the MICs of the FEA are much lower than that of the AME against all susceptible strains, confirming that the antimicrobial activity of *A. majus* may be due to the endophytic metabolites. The ethanol extract of *A. majus* was reported to inhibit different species of *Staphylococcus* and *Streptococcus* bacteria with MIC ranging from 3.9–15.6 mg/mL and exhibited stronger antibiofilm activity against *Staphylococcus* species than *Streptococcus* species, however, none of the extracts used in that study was able to inhibit biofilm formation completely [18].

In this study, fractionation was done on the FEA to explore the most active fraction and the promising metabolites responsible for these activities. Interestingly when testing the three fractions for antimicrobial and antibiofilm activities, they showed more potent activity than FEA with fraction III being the strongest fraction in either inhibiting the microorganisms or disrupting the preformed biofilms. Purification of fraction III yielded a major compound **1**; dihydroauroglaucin. This pure compound revealed the best antimicrobial results against all types of tested microorganisms. More interesting that both AME and FEA extracts showed very low cytotoxicity against normal human fibroblastic cell line BHK while compound **1** dihydroauroglaucin revealed only moderate cytotoxicity at the highest used dose, revealing good cytocompatibility with human normal cell lines [7]

The fungal extract fractions were found to contain active secondary metabolites as prenylated benzaldehyde derivatives namely; dihydroauroglaucin (1), tetrahydroauroglaucin (2), 2-(-3,6-dihydroxy-hepta-1,4-dien-1-yl)-3,6-dihydroxy-5-(dimethylallyl)benzaldehyde (3), isotetrahydroauroglaucin (4), and flavoglaucin (5). These compounds were previously isolated from different *Aspergillus* sp. [23,24,41].

Bacterial resistance in their free-living forms is a main concern for the health system, however, when they are present in biofilm, the problem is even aggravated [42]. National institutes of health announced that more than 75% of microbial infections that occur in the human body are corroborated by the development and firmness of biofilms. In most biofilms, microorganisms account for less than 10% of the dry mass, whereas the matrix can account for over than 90%. The matrix consists of hydrated extracellular polymeric materials mostly produced by the organisms themselves, in which the biofilm cells are embedded. This matrix provides the mechanical rigidity of biofilms, facilitates their adhesion to surfaces, and forms a consistent three-dimensional polymer network that interconnects and quickly immobilizes the biofilm [43], thus, established biofilms require 10–1000 times higher concentrations of antimicrobial agents and are extremely reluctant to phagocytosis, making biofilms very difficult to eliminate from living hosts [44]. 

Biofilm forming microorganisms are implicated in many infective diseases such as otitis media, periodontitis, dental caries, and osteomyelitis and in chronic diseases such as pulmonary infections of cystic fibrosis patients [16,45]. A survey conducted in 2012, included around 2000 healthcare facilities and 300,000 patients revealed that most nosocomial infections are highly associated with strong biofilm-producing *E. coli*, *Staphylococcus* sp., *P. aeruginosa*, and *K. pneumonia* [46,47]. In this study, dihydroauroglaucin was significantly effective in disrupting the 24 h preformed biofilm with (MBIC = 31.25 µg/mL) for all tested biofilm forming microorganisms. This concentration is much lower than that of fraction III (125 µg/mL) or the total fungus extract (250 µg/mL) by 2- and 4-folds, respectively. This finding is very promising in searching for a broad-spectrum antimicrobial agent with potent effect on established biofilm that may be used alone or in combination with antibiotics as this compound can interfere with the biofilm matrix allowing the antibiotic to reach the bacterial cells. In that context, Lemos et al., reported the synergistic effect between psychorubrin isolated from *Mitracarpus frigidus* and chloramphenicol in inhibition of biofilm of *S. aureus* and *Streptococcus pyogenes* [42].

*S. mutans* is a primary etiologic agent of human dental caries by its ability to form biofilms on the hard tissues of the human oral cavity [48], while *S. aureus* biofilm associated with implanted medical devices are difficult to treat and result in the replacement of these devices [49,50]. Interestingly in the present study the MBIC of dihydroauroglaucin against *S. mutans* and *S. aureus* biofilms was 15.6 and 7.8 µg/mL, respectively. 

*C. albicans* is the fourth most frequent cause of nosocomial bloodstream infections and are the major fungal species segregated from medical device infections [51]. Mechanical heart valves, pacemakers, urinary and central venous catheters, contact lenses, joint prostheses, and dentures are all susceptible to *C. albicans* biofilms. Since fungal biofilms are generally resistant to existing antifungal drugs, high antifungal dosages with elimination of the colonized medical tools are usually obligatory to cure infections with the risk of higher doses toxicity or removal of serious devices such as artificial heart valves and joints [51]. Accordingly, the potent *C. albicans* antibiofilm activity of dihydroauroglaucin compound isolated in this study may play an important role in the treatment.

## 4. Materials and Methods

### 4.1. Plant Material

*A. majus* fruits were collected in November 2017 from Arab Company of Pharmaceuticals and Medicinal Plants (Mepaco-Medifood) El-Sharkya, Egypt with coordinates near (30.3799° N, 31.4544° E). They were preserved in well closed containers at room temperature. Plant samples were identified by Dr. Mokhtar Bishr Technical Director of Mepaco Company and stored in a herbarium with the number RD-235-018.

### 4.2. Extraction of the Plant Material

The air-dried fruits of *A. majus* (500 g) were reduced to coarse powder and extracted with ethanol 70% at room temperature till exhaustion. The ethanolic extract was concentrated by evaporation under reduced pressure at 60 °C to give a sticky dark brown extract named as AME which was stored in amber glass well closed container in a refrigerator till usage.

### 4.3. Endophytic Fungi Isolation

Endophytic fungi were segregated by the method described by Hazalin et al. [52]. Concisely, *A. majus* fruits were washed with sterile water, sterilized with 70% ethanol for 1 min followed by rinsing twice with sterile water. The dried fruits were crushed and aseptically placed in Potato Dextrose Agar (PDA) (Oxoid, Hampshire, UK) plates supplemented with 250 mg/L of both streptomycin and gentamicin to inhibit bacterial growth. Non-crushed, surface sterilized fruits were also cultured to exclude the presence of epiphytic fungi, in addition to non-inoculated PDA plates which served as negative control. The plates were incubated for seven to 14 days at 25 °C. Various mycelia growing out of the segments were cultivated and the isolated pure fungi were maintained on PDA slants. This experiment was repeated three times.

### 4.4. Fungus Morphology and Microscopic Observation

Morphological identification of the isolated fungus was carried out according to the standard taxonomic key including colony characters such as texture, shape, and color [53]. Prospective fungus was cultured using slide culture method for seven days on PDA [54], then the mycelia were detected under microscope after addition of lactophenol cotton blue. The morphological characters of hyphae and conidia were used for fungus identification.

### 4.5. Fungus Identification through Molecular Approach

Genomic DNA was extracted according to Sigma Scientific Services Co. ITS 1 (5′-TCC GTA GGT GAA CCT GCG G-3′) and ITS 4 (5′-TCC TCC GCT TAT TGA TAT GC-3′) were used as forward and reverse primers respectively to amplify ribosomal ITS region. Thermal cycling conditions were as follows: Initial denaturation at 95 °C for 10 min, 35 cycles of denaturation at 95 °C for 30 s, annealing at 57 °C for 1 min, and extension at 72 °C for 1.5 min. The post cycling expansion was done as one cycle at 72 °C for 10 min. 

The PCR yields were then purified by GeneJET PCR Purification Kit (Thermo K0701, Waltham, MA, USA) in accordance with to the manufacturer’s directions and the refined DNA was stored at –20 °C. Finally, the refined PCR product was sequenced using ABI 3730xl DNA sequencer. The final sequence of the gene product of the fungus isolate was aligned against available sequences in GenBank database using NCBI BLAST (Basic Local Alignment Search Tool; http://blast.ncbi.nlm.nih.gov/). The neighbor-joining technique was used to build the phylogenetic tree using the MEGA 5 software. The identified isolate sequence was deposited to GenBank database and assigned an accession number [55,56]. 

### 4.6. Solid State Fermentation and Extraction of the Fungus Metabolites

Solid rice media for mass production was prepared by adding 100 g of rice mixed with 120 mL of sterilized water in 1L Erlenmeyer flasks sealed with cotton and autoclaved at 121 °C for 20 min. Plugs from PDA fungal cultures were used to inoculate fifteen solid rice flasks and they were allowed to grow for 21 days at room temperature. Fungal metabolites were extracted till exhaustion using ethyl acetate (EtOAc) (3 × 600 mL) as described [25], and then evaporated under vacuum and the dark reddish-brown residue of FEA was reserved for biological and chemical investigation.

### 4.7. Antimicrobial Susceptibility Assay for AME and FEA Extracts 

A rapid screening of the antibacterial and antifungal activity for AME and FEA extracts was done to confirm their previously reported activity [18,32,57]. Colorimetric broth micro-dilution method using XTT [2,3-bis(2-methoxy-4-nitro-5-sulfo-phenyl)-2*H*-tetrazolium-5-carboxanilide]-reduction assay was adopted to determine MIC against Gram positive bacteria: *S. aureus* (ATCC 25923) and *S. mutans* (ATCC 25175), as well as Gram negative bacteria: *E. coli* (ATCC 25922) and *P. aeruginosa* (ATCC9027). In addition to *A. fumigates* (ATCC MYA4609) and *C. albicans* (ATCC 10231) representing mold and yeast, respectively [58,59]. Before each experiment, bacterial and fungal strains were cultured overnight at 37 °C and 25 °C in Tryptone Soya Broth (TSB) and Sabouraud Dextrose Broth (SDB) media, (Oxoid, UK), respectively. XTT (Sigma) was prepared in a saturated solution at 0.5 g/L in Ringer’s lactate. The solution was sterilized through a 0.22-μm-pore-size filter, aliquoted, and stored at −70 °C. Prior to each assay, an aliquot of stock XTT was thawed, and menadione (Sigma; 10 mM prepared in acetone) was added to a final concentration of 1 μM. The extract was serially diluted in DMSO, and then 50 µL of each dilution were added to wells in microtiter plate containing 100 µL TSB for bacterial strains or 100 µL SDB for fungi. Fifty µL of adjusted microbial inoculum (10^6^ cell/mL) was added to each well, and then the microtiter plates were incubated in the dark at 37 °C for 24 h for bacteria, 48 h, and four days at 25 °C for yeast and fungi, respectively. After incubation, 100 µL of freshly prepared XTT were added, incubated again for 1 h at 37 °C. Colorimetric variation in the XTT assay was measured using a microtiter plate reader at 492 nm. The MIC was specified as the extract concentration that produced 100% decrease in optical density compared with control growth results. Ciprofloxacin and amphotericin B were used as standard antibacterial and antifungal agents, respectively. 

### 4.8. Biofilm Inhibition Assay Using AME and FEA Extracts

The effect of both AME and FEA extracts on established biofilms was tested by static microtiter plate method according to Lemos et al. [42,60]. Briefly, the biofilm-forming microorganisms: *S. aureus* (ATCC 25923), *S. mutans* (ATCC 25175), *E. coli* (ATCC 25922), and *P. aeruginosa* (ATCC 9027) were cultured at 37 °C in TSB for 18 h and *C. albicans* (ATCC 10231) was cultured at 37 °C for 48 h. Biofilm cells were produced with each of these organisms using 96-well polystyrene microtiter plates loaded with 180 µL TSB for all bacterial strains and SDB for *C. albicans* and 20 µL of 10^7^ cells/mL inoculums for bacteria and 10^6^ cells/mL for yeast. The microtiter plates were incubated for 24 h at 37 °C. After incubation, the planktonic cells were gently discarded, wells were washed three times with phosphate buffer saline (PBS), and then 200 µL of sterile media was added to each well, followed by twofold serial dilution of the extract. The microtiter plates were re-incubated at 37 °C for 24 h. After incubation, the planktonic cells were gently removed, and the remaining biofilm cells were rinsed three times with distilled water and stained with 0.01% crystal violet for 15 min. The plates were washed again with distilled water to remove the excess stain, resuspended in 70% ethanol for 10 min. Optical density (OD) was determined at 590 nm. The amount of biofilm inhibition was calculated relative to the amount of biofilm that was grown in the absence of the extract (defined as 100% biofilm) and the media sterility control (defined as 0% biofilm). All experiments were performed in triplicate. 

### 4.9. Fractionation and Purification of FEA Extract Compounds

To investigate the antimicrobial and antibiofilm activity based on bio-guided fractionation, the FEA (25 g) was applied on vacuum liquid chromatography (VLC) packed with silica gel 60 mesh (Merck, Darmstadt, Germany), elution was performed using a gradient system of *n*-hexane-EtOAc from 100 to 0% followed by DCM-MeOH from 100 to 0%. The collected fractions were investigated using thin layer chromatography plates by UV different wave lengths (254 and 365nm) and sprayed with vanillin/sulfuric acid reagent. Each fraction was further clarified using preparative HPLC (C-18 column) eluted with gradient acetonitrile-H_2_O (90–100%). Last purification steps were achieved by means of preparative HPLC (Knauer, Berlin, Germany) on Kromasil ODS preparative column (10 by 250 mm) with flow rate 4 mL/min and UV detection. The identification and elucidation of the isolated compounds from the fractions were done using NMR spectra recorded on a Bruker AVANCE HD III 400 MHz spectrometer (Bruker, Fällanden, Switzerland). A schematic fractionation protocol was shown in Appendix A.

### 4.10. Antimicrobial Susceptibility and Biofilm Inhibition Assay Using FEA Fractions and Purified Compound

The fractions of fungal extract and the major compound of the highly active fraction were assayed for their antimicrobial and antibiofilm activities as previously described.

### 4.11. Cell Viability Determination

Cytocompatibility of the AME, FEA and dihydroauroglaucin compound was evaluated on normal human fibroblastic cell lines using the Sulphorhodamine-B (SRB) assay [61]. Roswell Park Memorial Institute 1640 (RPMI-1640) medium, SRB, BHK, were obtained from the American Type Culture Collection (Manassas, USA). They were maintained and grown at the Egyptian National Cancer Institute (Cairo, Egypt) in RPMI-1640 supplemented with 10% fetal bovine serum, 2 mM L-glutamine, 1.5 g/L sodium bicarbonate and 1% penicillin/streptomycin and incubated at 37 °C in 5% carbon dioxide (CO_2_). Samples were freshly dissolved in DMSO and then diluted with RPMI-1640 incubation medium before each experiment, so that the final concentration of DMSO was not more than 0.1% (*v*/*v*).

Briefly, exponentially growing normal human fibroblast cells were seeded in 96-well microtiter plates at an initial density of 5 × 10^3^/well. After 24 h, the samples were added to each well at various concentrations (ranging from 5–50 μg/mL) and incubated in a humidified, 5% CO_2_ incubator at 37 °C for 48 and 72 h. After incubation they were fixed with 10% trichloroacetic acid for 1 h at 4 °C and stained with 0.4% SRB for 30 min. The wells were then washed four times with 1% acetic acid, air-dried and the dye was solubilized with 10 mM Tris base (pH 10.5). The OD was measured spectrophotometrically at 570 nm with the microplate reader. The experiment was repeated three times. The percentage of cell survival was calculated as follows: Survival fraction ¼ O.D. (treated cells)/O.D. (control cells).

Cells cultured with DMSO plus medium alone were counted as control; results were conveyed as % viability relative to 100% of controls. 

### 4.12. Statistical Analysis

All statistics have been conducted using GraphPad Prism 5 (LaJolla, CA, USA) software. One-way analysis of variance (ANOVA) with Tukey’s Multiple Comparison post hoc test was utilized to calculate the significant difference between means. *p* < 0.05 was considered statistically significant.

## 5. Conclusions

The extensive failure of antibiotic treatment to eradicate biofilm-associated infections has enhanced the exploration of alternate curative agents. Endophytes are considered a hidden treasure that is required to be explored to find new and promising secondary metabolites with different activities. *A. majus* is a wild endemic plant in Egypt that has much pharmaceutical potential and the medicinal features of this plant may be a result of the ability of its endophytic fungi to produce effective secondary metabolites. Prenylated benzaldehyde derivatives are a class of compounds that were purified from *A. amstelodami* isolated from *A. majus*, L. fruits with versatile usage in medicine. In this study dihydroauroglaucin; a prenylated benzaldehyde derivative; proved to have broad spectrum antimicrobial and antibiofilm activities. More studies are required to investigate the activity of prenylated benzaldehyde derivatives on clinical biofilm forming strains and to evaluate the effect of combinations of the prenylated benzaldehyde with antibiotics to overcome antimicrobial resistance. Furthermore, studies are required to search for an easier, natural, more economic and valuable antimicrobial pharmaceutical products.

## Figures and Tables

**Figure 1 molecules-24-04118-f001:**
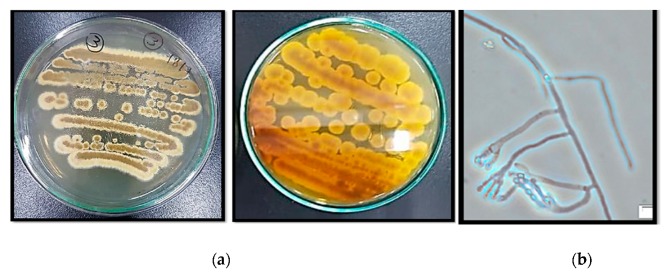
Morphological (**a**) and microscopical (**b**) characters of *A. amstelodami*.

**Figure 2 molecules-24-04118-f002:**
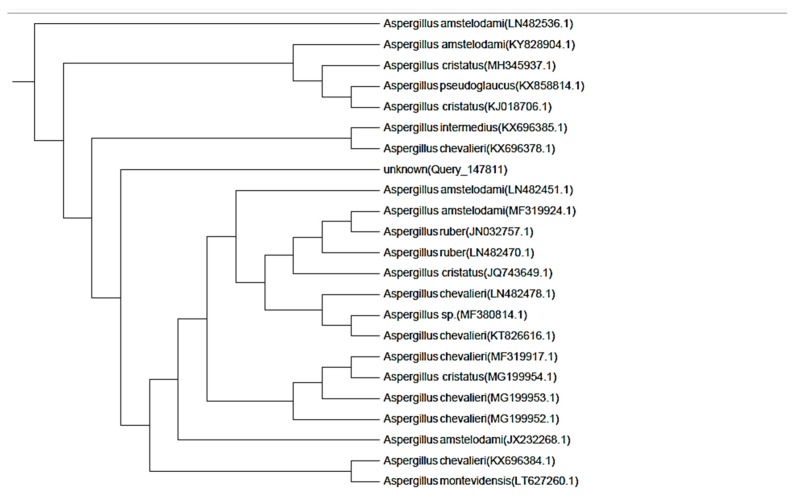
Phylogenetic tree of the isolated endophyte *A. amstelodami*.

**Figure 3 molecules-24-04118-f003:**
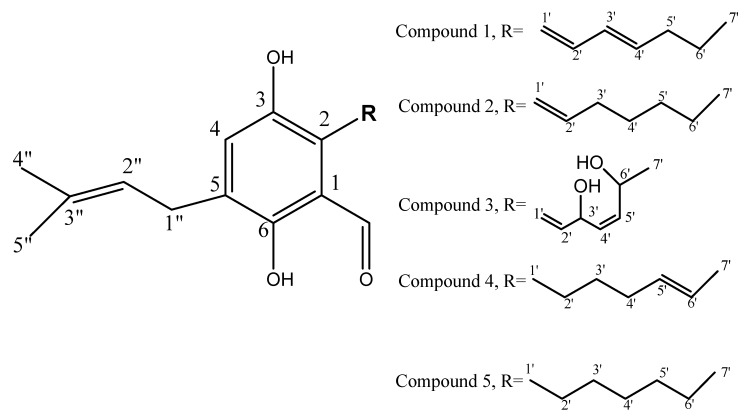
Compounds isolated from the ethyl acetate extract of *A. amsetoldami*.

**Figure 4 molecules-24-04118-f004:**
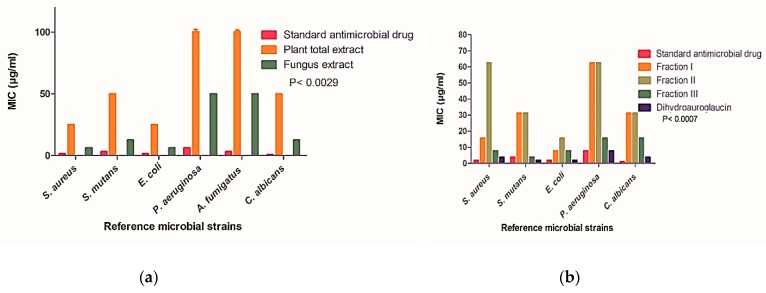
Minimum inhibitory concentration (MIC) values against different microbial reference strains. (**a**) MIC values of the *A. majus* ethanol fruits (AME) and the fungal ethyl acetate (FEA) extracts compared to standard ciprofloxacin and amphotericin B. (**b**) MIC values of FEA extract fractions and dihydroauroglaucin compared to standard ciprofloxacin and amphotericin B. All determinations were carried out in triplicate manner and values are expressed as means ± SD.

**Figure 5 molecules-24-04118-f005:**
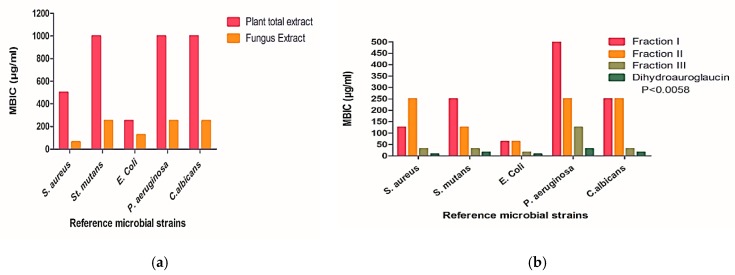
Antibiofilm activity of different fractions represented by minimum biofilm inhibitory concentration (MBIC) values. (**a**) The MBIC values of the *A. majus* ethanol fruits (AME) and the fungal ethyl acetate (FEA) extracts, (**b**) The MBIC values of FEA fractions I, II, and III and dihydrauroglaucin. All determinations were carried out in triplicates and values are expressed as the means ± SD.

**Figure 6 molecules-24-04118-f006:**
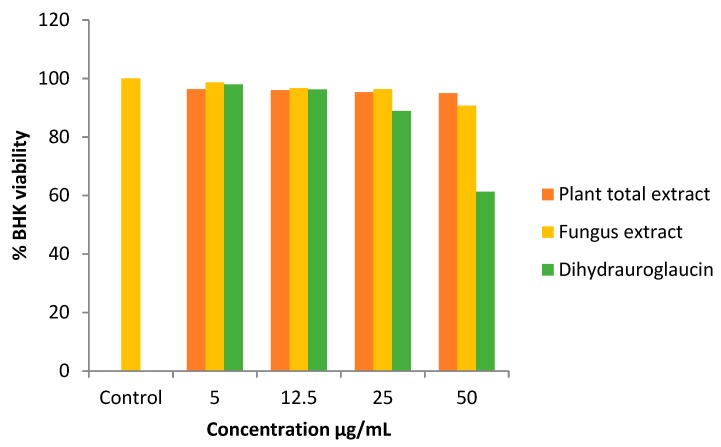
In-vitro cytocompatibility assay. Sulforhodamine B colorimetric assay for cytotoxicity screening evaluated on normal human fibroblast cell lines (BHK). All determinations were carried out in triplicates and values are expressed as the means ± SD.

**Table 1 molecules-24-04118-t001:** Morphological and microscopical description of *A. amstelodami*.

Morphological Characters		Microscopic Characters	
Surface	Yellowish Green	Hyphae	Branched septate
Margins	Entire	Conidia	Yellowish green (4 to 7 μm), roughened
Reverse side	Brownish yellow	Phialides	Single series (Uniseriate) covering nearly the entire vesicle
Growth	Slow to moderate		
Elevations	Umbonate		

**Table 2 molecules-24-04118-t002:** ^1^H-NMR and ^13^C-NMR/APT-NMR of compounds isolated from *A. amstelodami*.

	1		2		3	4	5	
Atom no	δ_H,_^1^H-NMR	δ_C_^13^C-NMR/APT	δ_H,_ ^1^H-NMR	δ_C_^13^C-NMR/APT	δ_H,_ ^1^H-NMR	δ_H,_ ^1^H-NMR	δ_H,_ ^1^H-NMR	δ_C_^13^C-NMR/APT
1	-	130.8 C	-	117.5 C	-	-	-	115 C
2	-	127.5 C	-	123.8 C	-	-	-	128.8 C
3	-	145.5 C	-	145.2 C	-	-	-	144.8 C
4	7.0 (s), 1H	125.7 CH	7.04 (s), 1H	125.3 CH	7.02 (s), 1H	6.36 (m), 1H	6.9 (s), 1H 4.6 (s), OH	125.4 CH
5	-	135.8 C	-	130.3 C	-	-	-	133.6 C
6	11.75 (s), OH	158.5 C	11.77 (s)OH	155.6 C	11.95 (s), OH	11.83 (s), 1H	11.95 (s), OH	154.8 C
7	10.1 (s)	196.5 CH	10.2 (s), 1H	196.7 CH	10.3 (s), 1H	10.12 (S), 1H	10.3 (s), 1H	195.9 CH
1′	6.59 (d, *J* = 15.8 Hz, 1H)	140.7 CH	6.50 (d, *J* = 16.1 Hz, 1H)	121.05 CH	6.92 (d, *J* = 5.5 Hz, 1H)	2.99 (m), 2H	2.9 (m), 2H	24.7 CH_2_
2′	6.46 (dd, *J* = 15.8, 10.1 Hz, 1H)	132.19 CH	6 (m). 1H	142.8 CH	6 (m), 1H	1.6 (m), 2H	1.6 (m), 2H	32.07 CH_2_
3′	6.3 (m), 1H	125.49 CH	2.3 (m), 2H	34.03 CH_2_	5.3 (m), 1H	1.4 (m),2H	1.4 (m), 2H	29.7 CH_2_
4′	5.47 (m), 1H	119.9 CH	1.5 (m), 2H	28.9 CH_2_	6.1 (m), 1H	1.3 (m), 1H	1.31 (m), 2H	29.1 CH_2_
5′	2.1 (m), 2H	34.9 CH_2_	1.4 (m), 2H	31.47 CH_2_	6.1 (m), 2H	5.3 (m),1H	1.38 (m), 2H	31.7 CH_2_
6′	1.3 (m), 2H	29.9 CH_2_	1.3 (m), 2H	22.2 CH_2_	1.29 (m), 1H	5.4 (m), 1H	1.29 (m), 2H	22.7 CH_2_
7′	0.9 (m), 3H	13.40 CH_3_	0.94 (m), 3H	14.3 CH_3_	1.3 (m), 3H	1.86 (d, *J* = 1.6 Hz, 1H)	0.9 (m), 3H	14 CH_3_
1″	3.34 (d, *J* = 7.8 Hz, 1H)	27.1 CH_2_	3.34 (d, *J* = 7.4 Hz, 2H)	27.18 CH_3_	3.32 (d, *J* = 7.4 Hz, 1H)	3.44 (d, *J* = 7.6 Hz, 1H)	3.31 (d, *J* = 7.5 Hz, 1H)	27.02 CH_2_
2″	5.3 (m), 1H	121.2 CH	5.3 (m), 1H	120.5 CH	5.3 (m), 1H	5.88 (m), 1H	5.3 (m), 1H	121.2 CH
3″	-	139.8 CH	-	133.8 C	-	-	-	133.9 C
4″	1.7 (s), 3H	17.7 CH_3_	1.7 (s), 3H	18.29 CH_3_	1.7 (s), 3H	1.75 (s), 3H	1.7 (s), 3H	17.7 CH_3_
5″	1.8 (s), 3H	25.9 CH_3_	1.8 (s), 3H	25.7 CH_3_	1.8 (s), 3H	1.85 (s), 3H	1.8 (s), 3H	25.7 CH_3_

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
