# Peer review of "Bio-Guided Fractionation of Prenylated Benzaldehyde Derivatives as Potent Antimicrobial and Antibiofilm from Ammi majus L. Fruits-Associated Aspergillus amstelodami"

_molecules, 2019, doi:10.3390/molecules24224118_

Round 1
Reviewer 1 Report
This is a very interesting paper, which aims at finding a novel compound with antibacterial activity, from a natural source. They isolated the endophytic fungi associated with A. majusfruits and verified the antimicrobial and antibiofilm activity.
Introduction
I would suggest to briefly provide an overview on the plant-derived compounds proposed in literature as alternatives to traditional antimicrobials, to emphatize the role of natural agents for selectives new antibacterial molecules.
Essential oils in particular are receving a growing interest; please include this topic with references:
https://www.ncbi.nlm.nih.gov/pubmed/27650979
https://www.ncbi.nlm.nih.gov/pubmed/26007187
MM and results
It would be interesting to have also some data about cytocompatibility.
Discussion
I would suggest to add some recent references on the use of compounds derived from endophytic fungi with antibacterial activity, such as:
https://www.ncbi.nlm.nih.gov/pubmed/31617784
https://www.ncbi.nlm.nih.gov/pubmed/31602502
Minor grammar errors can be found along the text.
Author Response
First of all, we would like to thank all reviewers for their supporting words and helpful suggestions.
Response to reviewer 1 comments for our manuscript molecules-627908
|
Serial
|
Comment |
Response |
|
1 |
Introduction I would suggest to briefly provide an overview on the plant-derived compounds proposed in literature as alternatives to traditional antimicrobials, to emphatize the role of natural agents for selectives new antibacterial molecules. |
Thanks for your suggestion and it was added in the introduction. |
|
2 |
Introduction Essential oils in particular are receving a growing interest; please include this topic with references: |
Introduction was modified to include this part. |
|
3 |
MM and results It would be interesting to have also some data about cytocompatibility.
|
Cytocompatibility of total plant extract, total fungal extract and the purified promising compound were added to the manuscript as we already did this experiment using normal cell line in addition to cancer cells and we keep searching and investigating this point as it gives promising preliminary data (to be published after complete investigations). |
|
4 |
Discussion I would suggest to add some recent references on the use of compounds derived from endophytic fungi with antibacterial activity,
|
Recent references were added to the discussion. |
|
5 |
Minor grammar errors can be found along the text. |
The manuscript checked for grammar mistakes |
Reviewer 2 Report
Ammi majus is known to contain biologically active compounds so it has much pharmaceutical potential. It has been previously reported that A. majus fruit extract possesses antimicrobial activity. In this manuscript, the authors report for the first time the isolation of an endophytic fungus, Aspergillus amstelodami, associated with Ammi majus fruits. A bio-guided fractionation method of fungal extract resulted in production of three fractions and 5 pure compounds. The authors carry out a complete and very interesting study of the antimicrobial and antibiofilm activity of the fruits and fungal extract and of the major isolated compound, dihydroauroglaucin, and found that fungal extract and the compound displayed the highest antimicrobial activity. Therefore, this work suggests that the medicinal features of this plant may be a result of the ability of its endophytic fungi to produce effective secondary metabolites.
The studies are designed and conducted in a logical manner and all conclusions match the data presented. A few points should be addressed prior to publication.
-The Abstract needs a statement at the end with the suggestion that the medicinal features of this plant may be a result of the ability of its endophytic fungi to produce effective secondary metabolites. This seems to be the overall conclusion of the data presented.
-Line 80, 111, 184, 186 and 224 Abbreviations should be explained in the main text: ITS rRNA, APT, AME, FEA, MIC and MBIC.
-Pay attention to spaces when words written in italics appear all over the paper.
Author Response
First of all we would like to thank all the reviewers for their supporting words and valuable suggestions.
Response to reviewer 2 comments for our manuscript molecules-627908
|
Serial |
Comment |
Response |
|
1 |
The studies are designed and conducted in a logical manner and all conclusions match the data presented. |
We would like to thank you for supportive words and comments. |
|
2 |
The Abstract needs a statement at the end with the suggestion that the medicinal features of this plant may be a result of the ability of its endophytic fungi to produce effective secondary metabolites. This seems to be the overall conclusion of the data presented. |
Thanks for your suggestion and a statement was added to conclude all the results. |
|
3 |
Line 80, 111, 184, 186 and 224 Abbreviations should be explained in the main text: ITS rRNA, APT, AME, FEA, MIC and MBIC. |
All missed abbreviations were checked and added. |
|
4 |
Pay attention to spaces when words written in italics appear all over the paper. |
The manuscript checked again for correct spacing between words. |